# Improving Intensive End-of-Life Care for Infants and Children: A Scoping Review of Intervention Elements

**DOI:** 10.3390/children12111485

**Published:** 2025-11-03

**Authors:** Elizabeth G. Broden Arciprete, Na Ouyang, Sarah E. Wawrzynski, Ijeoma J. Eche-Ugwu, Janene Batten, Deena K. Costa, Shelli L. Feder, Jennifer M. Snaman

**Affiliations:** 1National Clinician Scholars Program, School of Medicine, Yale University, New Haven, CT 06520, USA; 2School of Public Health, Yale University, New Haven, CT 06510, USA; 3School of Nursing, Yale University, Orange, CT 06477, USA; 4Center for Healthcare Delivery Science, Nemours Children’s Health, Delaware Valley, DE 19803, USA; 5Phyllis F. Cantor Center for Research in Nursing and Patient Care Services, Dana-Farber Cancer Institute, Boston, MA 02215, USA; 6Harvard Medical School, Boston, MA 02215, USA; 7Harvey Cushing/John Hay Whitney Medical Library, Yale University, New Haven, CT 06520, USA; 8Department of Supportive Oncology, Dana-Farber Cancer Institute, Boston, MA 02215, USA

**Keywords:** pediatric critical care, neonatal critical care, interdisciplinary, end-of-life care, palliative care

## Abstract

**Highlights:**

**What are the main findings?**
Most interventions targeted clinician knowledge as a primary outcome, whereas fewer interventions targeted family outcomes.Few interventions utilized nursing workflows to improve end-of-life care in pediatric and neonatal critical care settings.

**What are the implications of the main findings?**
Findings underscore the need for interventions that target family outcomes, especially parental empowerment.More interventions should assess family outcomes and aim to integrate families in development.

**Abstract:**

Background/objectives: High-quality pediatric critical care includes supporting children nearing the end-of-life (EOL) and their families. Cataloging existing interventions for children dying in the neonatal or pediatric intensive care unit (NICU, PICU) establishes critical areas for future research. In this scoping review, we evaluated characteristics of PICU EOL interventions. Methods: A librarian guided a search of OVID Medline, CINAHL, OVID PsycINFO, OVID Embase, Cochrane Central, and Web of Science, plus backwards and forwards reference searching. We included interprofessional interventions, defined as any systematic change (e.g., educational programs, symptom management, electronic medical record, etc.), for children dying from any cause. Studies were independently screened by two reviewers. Data were extracted by one team member and reviewed by a second. We extracted intervention elements, contextual factors, implementation barriers/facilitators, and generated frequencies from qualitative coding. Results: Of 11,643 screened articles, 44 met the inclusion criteria. Most were in neonatal ICUs (n = 28/44, 64%) and general PICUs (n = 10/44, 23%). Most interventions aimed to improve clinician knowledge (25/44, 57%), augment clinical structures and processes (n = 11/44, 25%), or enhance communication (n = 8/44, 18%). Common delivery methods included clinical practice changes (n = 25/44, 57%; e.g., protocols, order sets [n = 12]), and educational sessions (n = 20/44, 45%). Outcomes included clinician knowledge (n = 17/44, 39%), qualitative feedback (n = 18/44, 41%), feasibility/acceptability (n = 12/44, 27%), or treatment utilization (n = 11/44, 25%). Few examined families’ mental health (n = 3, 7%) or bereavement (n = 2, 5%). Few reported implementation facilitators or barriers. Conclusions: Most included studies targeted clinician outcomes through education. Designing, testing, and implementing interventions focused on family outcomes is a critical next step.

## 1. Introduction

A child’s death is life-altering, leading to life-long grief for their families [1,2,3,4,5,6,7]. Many children and families navigate end-of-life (EOL), the last days or hours of an illness or injury, in a neonatal or pediatric intensive care unit (NICU, PICU) [8,9,10]. Optimal EOL care for infants and children care includes a multidisciplinary, team-based approach that considers physical symptoms, family spiritual and emotional needs, and meaning-making opportunities [7,11,12]. As the clinicians most frequently at the bedside, nurses are well-positioned to assess, evaluate, and connect families with the support they need during EOL. Interprofessional EOL care that fully leverages nurses’ roles is therefore critical to supporting families and minimizing adverse grief outcomes.

Optimizing interprofessional EOL care for infants and children in the ICU setting requires an evidence base of rigorously tested interventions. Generally low pediatric mortality rates mean that clinicians may rarely apply palliative and EOL skills learned during pre-licensure training, instead learning on the job. Studies of family and clinician experiences illuminate several gaps in EOL care that exacerbate parental distress [13,14,15,16]. These include fragmented communication, strained relationships within and between clinical teams and families, and limited structural support for families and clinicians, among others [13,14,15,16]. Supportive interventions that can be readily integrated into routine care could help address these known barriers to high-quality EOL care. While many interventions focused on improving EOL care have been developed, a systematic understanding of what elements interventions include and which parent-identified gaps they target is lacking.

Although EOL care interventions are increasingly common, systematic evaluations of their components—including who, how, and when interventions are delivered and implemented—remain lacking. Systematically cataloging existing interventions is necessary to ensure all dying children in the NICU/PICU receive compassionate, evidence-based care from well-trained clinicians. Therefore, we conducted a scoping review to evaluate interventions for children nearing EOL in the NICU/PICU.

## 2. Materials and Methods

### 2.1. Design

This scoping review [17] followed Arksey and O’Malley’s method [18], including (1) identifying the research question (Section 2.1), (2) identifying relevant studies (Section 2.2), (3) study selection (Section 2.3), (4) charting the data (Section 2.5 and Section 2.6), and (5) collating, summarizing, and reporting the results (Section 2.6 and Section 3). We did not incorporate a patient and family engagement step as proposed by Levac et al. [19]. The following research question guided the review: how are pediatric EOL interventions designed and delivered, who delivers and receives them, what outcomes do they aim to effect, and what factors facilitate or inhibit their implementation?

We used the Preferred Reporting Items for Systematic Reviews and Meta-Analyses (PRISMA-ScR) guidelines Extension for Scoping Reviews [20]. The review protocol was registered on the Open Science Foundation (https://doi.org/10.17605/OSF.IO/7ST2E accessed on 1 October 2025). We made minor protocol amendments, specifically to the synthesis and analysis phase, as needed based on the content of included articles. The manuscript was published on a pre-print server on 6 October 2025 (https://doi.org/10.20944/preprints202510.0396.v1, accessed on 1 October 2025).

### 2.2. Search Strategy

A medical librarian (JB) and the first author (EGBA) developed a search strategy that was peer-reviewed by another medical librarian (Appendix A). Electronic databases included Ovid MEDLINE(R) ALL, OVID APA PsycINFO, OVID Embase, CINAHL, Cochrane Central, and Web of Science. Search terms included both controlled vocabulary and synonymous free text to capture these concepts: population (children with life-threatening conditions), care settings (intensive care units), and interventions (intervention, education, training, etc.). We used Citationchaser [21] to identify additional articles using backward and forward reference searching.

### 2.3. Selection Criteria

To capture interprofessional interventions, articles with the search terms in the title, abstract, or subheadings and the word “nurse” within the full text were included. This allowed us to evaluate interventions that meaningfully involved nurses in their implementation and impact in a variety of ways. We included reports of empirically evaluated interventions designed to support EOL care for children (i.e., last days of life) dying from any cause in any intensive care setting (e.g., neonatal, pediatric, and cardiac). We defined intervention broadly as any practice change focused on bedside EOL care (e.g., educational programs, symptom management, communication training, order sets, electronic medical record modifications, etc.). Eligible articles formally evaluated the interventions using quantitative, qualitative, or mixed methods, with outcomes reported by patients, families, or clinicians. We included all research designs and quality improvement efforts.

To capture as many interventions as possible, we made no exclusions based on publication year, location, or language. We translated articles published in languages other than English using Google translate [22] or ChatGPT (GPT-4; OpenAI) if needed. Studies were included if they reported results specific to ICU populations or if >50% of patients died in the ICU. Studies not explicitly focused on the EOL period were included only if >50% of patients died during the study period. We excluded reviews, opinion articles, dissertations, and editorials. Articles describing clinical practice changes without formal evaluation were excluded.

### 2.4. Screening and Data Management

Search results were uploaded and deduplicated in EndNote (version 20—Thompson Reutres) then uploaded into COVIDENCE systematic review software (Veritas Health Information) [23]. Two reviewers from a team of four authors (EGBA, NO, SEW, IJEU) independently screened study titles and abstracts. Discrepancies were resolved by a third reviewer from the team of four. Full text studies that met criteria were reviewed using the same process and team.

### 2.5. Data Extraction and Quality Appraisal

We developed and pilot-tested a standardized data extraction form in REDCap [24,25]. Three authors (EGBA, NO, or SEW) independently extracted article information and reviewed extracted data, with discrepancies resolved through discussion. We extracted intervention elements, contextual factors, and implementation barriers and facilitators (Appendix A). Intervention elements (Figure 1: Box 1) included development processes, goal or intended impact, delivery methods, interventionists and recipients of the intervention, and outcome measures (quantitative metrics and qualitative questions). Contextual factors included sample characteristics, ICU type (neonatal, general PICU, cardiac PICU), and hospital characteristics. We extracted unit- and hospital-level intervention implementation considerations based on the Consolidated Framework for Implementation Research, including barriers and facilitators within the domains of innovation design, outer setting, inner setting, and individuals [26,27]. We also extracted palliative care focus—physical, psychological, spiritual/existential, and/or social [28]. We evaluated methodological rigor using the Mixed Methods Appraisal Tool (MMAT; Appendix A) [29]; we did not evaluate quality improvement, program evaluation, or implementation project rigor given their small number and heterogeneous designs.

### 2.6. Analysis

Quantitative data were summarized in STATA [30] using frequencies for categorical and means/medians for continuous variables. For qualitative data, the coding team (EGBA, NO, SW, IJEU) generated descriptive categories from extracted free text about intervention elements and implementation barriers and facilitators from the CFIR [26,27,31], then collaboratively coded 10% of articles in Dedoose [32] before proceeding independently with all coding verified by the first author (EGB). Code frequencies describing intervention elements and implementation barriers and facilitators were generated. Categories were not mutually exclusive and, thus, do not sum to 100% in reporting.

To assess if interventions targeted parent priorities, we mapped intervention goals and outcomes to parent-identified gaps in EOL care. Based on two systematic reviews of parent PICU EOL experiences and one qualitative study of bereaved parents’ recommendations for PICU EOL care [13,16,33], we identified seven parent-identified gaps from these three articles: communication and information sharing, comfort and symptom management, family psychosocial and spiritual care, structural support (e.g., visitation policies), meaning-making (e.g., memento-making), grief and bereavement care, and parental role acknowledgement and empowerment. We tabulated the number of included articles addressing each of these gaps and intervention goals/outcomes.

## 3. Results

We screened 11,643 articles, assessed 286 full texts for eligibility, and 44 met inclusion criteria (Figure 2). Study designs included non-randomized quantitative (17/44, 39%) [34,35,36,37,38,39,40,41,42,43,44,45,46,47,48,49,50], quantitative descriptive designs (12/44, 27%) [51,52,53,54,55,56,57,58,59,60,61,62], qualitative (4/44, 9%) [63,64,65,66], mixed methods (3/44, 7%) [67,68,69], or other (e.g., quality improvement, pre-post, multimethod, etc.; 8/44, 18%) [70,71,72,73,74,75,76,77]; see Table 1. The neonatal ICU was the most common site (28/44, 64%) followed by general PICU (10/44, 23%). Four (9%) studies were conducted in pediatric cardiac ICUs. Most interventions focused on multiple palliative care domains (38/44, 86%).

### 3.1. Intervention Elements

Here we summarize intervention elements, contextual factors, implementation facilitators and barriers, and study rigor. Appendix A: The Table of Evidence contains study details and all citations.

Development: eleven out of forty-four (25%) interventions were developed from prior evidence (e.g., based on prior research, adapted existing interventions) and five (11%) were informed by clinical experience. Eighteen interventions (41%) engaged key partners (e.g., clinicians) in study development; three (7%) involved family members. For example, Zhang et al. engaged bereaved parents to develop a protocol for transferring infants to a specified NICU room with parent sleeping space [35]. Development processes were not explicitly reported in 21/44 (48%) articles.

Goals: Most (25/44, 57%) interventions focused on improving clinician knowledge or comfort with EOL care. Eleven (25%) interventions focused on improving clinical structures and processes. For example, Younge et al. integrated a palliative care protocol, order set, nursing care plan, and comfort medication guidelines into NICU EOL care [36]. Eight (18%) interventions aimed to enhance communication within clinical teams and with families, by increasing regular case discussions [37] or bereavement debriefings for families [38]. Six interventions (14%) integrated palliative care into the ICU; two (5%) supported decision making, [39,40] such as through meetings about discontinuation of life-sustaining treatments [40]. Nine (20%) interventions focused on staff debriefing. Interventions addressing patient and family experiences (12/44, 27%) included symptom management (n = 5, 11%), [36,41,42,43,44] memento-making activities (n = 4, 9%), [35,45,46,47] and support for parents (n = 3, 7%) [35,48,49].

Delivery: Most interventions were delivered through clinical practice changes (25/44, 57%). These included protocols/pathways (n = 12, 27%), team/family meeting processes/structures (n = 9, 20%), order set or medication administration instructions (n = 4, 9%), [36,41,42] electronic health records (n = 2, 5%), [50,51] or environmental modifications (n = 4, 9%) [35,48,49,52]. For example, Carter et al. incorporated EOL content into NICU morbidity and mortality conferences [53]. Other delivery methods included educational sessions (20/44, 45%) and simulations (n = 6, 14%). Ten of 44 (23%) interventions used reflective debriefings, such as peer support [54], emotional debriefing sessions for staff [52,55,56,57,58,59], or follow-up meetings with families [38,60]. Some interventions (7/44, 16%) were delivered by preparing specific personnel (e.g., champions, palliative care teams).

Timing and frequency of interventions: some interventions were integrated (n = 20/44, 45%) or separate (n = 24/44, 54%) from clinical processes. Four of forty-four (9%) interventions occurred once (e.g., singular memory-making experience) [35,45,46,47], and sixteen (36%) repeated changes within the practice setting (e.g., medication administration over the last days of a child’s life) [41]. Eleven (25%) interventions occurred once outside clinical practice (e.g., a one-time simulation) [61] and thirteen (30%) included multiple events outside clinical practice (e.g., simulations throughout PICU fellowship) [62]. Four interventions (9%) included post-mortem processes (e.g., a reflective debriefing, family bereavement meeting) [38,58,59,60].

Interventionists: About a quarter (24%) of interventionists included multiple disciplines (e.g., medicine, nursing, social work), and five (12%) used ICU-based teams. Three (7%) had nurse interventionists [41,48,63] and two (5%) interventionist teams included parents (Table 1) who completed intervention procedures such as photography [47] and educational content [64].

Subspecialty palliative care: Thirteen (39%) studies explicitly involved subspecialty palliative care in intervention development or delivery. In two (5%), no palliative care team was available [65,66], while three (7%) mentioned palliative care teams that were not part of the study [44,49,57]. Nineteen (43%) did not reference palliative care team involvement.

Summary of outcomes: Clinician knowledge, attitudes, or beliefs about palliative and EOL care practices were evaluated in 17/44 (39%) studies. Eighteen (41%) studies examined qualitative feedback. Twelve (27%) studies evaluated treatment utilization metrics (e.g., mechanical ventilation at time of death) [39]; twelve evaluated feasibility/acceptability (27%). Four (9%) evaluated medication use [36,42,43,65]. Five (11%) projects evaluated goals of care documentation or meetings [36,40,51,65,67], and/or patient symptoms/experience [41,42,43,44,67]. Three (7%) studies evaluated clinician mental health (e.g., burnout), [54,56,59] and three (7%) evaluated family mental health [35,39,44]. Two (5%) studies examined parent bereavement (e.g., meaning-making [38], grief experience [52]).

Mapping to parent-identified gaps: parent-identified gaps with the most corresponding interventions (Figure 3) were communication and information sharing (fourteen interventions) and comfort/symptom management (eleven interventions). Six interventions addressed family psychosocial and spiritual support, through integrating palliative care or improving clinical processes like psychosocial consultations. These were evaluated using treatment utilization metrics (n = 3) [35,51,68], or patient/family experience (e.g., parent depression/anxiety (n = 2)) [35,44]. Five interventions addressed structural support for parents through clinical structures/processes, addressing family experience (e.g., specific spaces for EOL) [35,49,52], or enhancing communication (e.g., increasing family conferences) [52]. Structural interventions evaluated family outcomes, treatment utilization metrics, and/or qualitative data.

Four interventions supported meaning-making through memento-making activities which were evaluated qualitatively [45,46,47,48]. Four interventions included grief and bereavement care, through integrating palliative care (n = 2) [39,69], supporting decision making (n = 1) [39], improving clinical structures (n = 1) [67], and/or enhancing communication [52]. For example, Morillo Palma et al.’s palliative care protocol included systematic bereavement follow-up for families [67]. Two interventions [35,48] helped support parental role empowerment by improving clinical structures and processes and addressing patient and family clinical experience, evaluated through family mental health outcomes and qualitative feedback. Specifically, Kymre et al. evaluated nurse-facilitated parent skin-to-skin opportunities for dying neonates [48].

### 3.2. Contextual Factors

Most studies (25/44, 57%) were conducted in academic institutions. Thirteen (30%) were in freestanding children’s hospitals and 6/44 (14%) in community or government hospitals. Six out of forty-four studies (14%) described multisite approaches.

### 3.3. Implementation Barriers and Facilitators

Hospital-level (outer setting) implementation facilitators included committed intervention champions (7/44, 16%), support for required time/resources (4/44. 9%) [59,68,70,72], established hospital EOL, palliative, or bereavement programming (7/44, 16%). Barriers included lack of programming (4/44, 9%) [50,65,69,70]. Many studies (26/44, 59%) did not report about hospital-level implementation factors.

Unit-level (inner setting) implementation facilitators included high volume (12/44, 27%), existing family-centered, palliative, or EOL care infrastructure (8/44, 18%). Individual level facilitators included invested individuals (8/44, 18%) and staff enthusiasm (2/44, 5%) [44,57]. Unit-level (inner setting) implementation barriers included technology or procedural issues (10/44, 23%), scheduling (10/44, 23%), and lack of palliative or EOL care infrastructure (3/44, 7%) [61,70,71]. Individual hesitations (2/44, 5%) [41,60] also functioned as barriers, for example, nurses who used an oral or transmucosal comfort medication protocol in Drolet et al.’s study reported discomfort interrupting families’ privacy [41].

### 3.4. Study Rigor

Common threats to rigor included potential selection or sampling bias (Appendix A: MMAT Results). Two-thirds, 66% (29/44), of studies reported demographic data of their sample. Most studies utilized a quasi-experimental design, often without a control group, introducing the potential for confounding.

## 4. Discussion

This scoping review cataloged EOL care intervention elements and implementation considerations in neonatal and pediatric ICUs. Most interventions aimed to improve clinician knowledge about palliative and EOL care through educational programming or improved clinical processes (e.g., team meeting procedures or assessment protocols). Few studies examined family outcomes. Parent-identified gaps of communication and comfort/symptom management were addressed by >10 interventions, whereas just two interventions addressed parental role empowerment. Implementation facilitators included structural factors like established palliative or bereavement programs. Barriers often included scheduling or operational challenges. These findings highlight common strategies used to enhance EOL care in the NICU/PICU and illuminate critical areas for future research and family engagement.

NICU and PICU nurses are uniquely positioned to help assess, support, and advocate for families’ dynamic needs during EOL. Over 40% of intervention teams involved nurses as part of the research team or in intervention design. However, just three (7%) studies evaluated nurse-led interventions that changed routine nursing practice, such as through hands-on care [66], assessment procedures [60], bereavement interventions, and follow up [70]. Given nurses’ uniquely close involvement with children and families during EOL, future interventions should consider leveraging nursing workflows, such as nurse communication and partnership with parents, to enhance EOL care.

Many studies support early palliative care during a child’s serious illness [78,79]. However, few studies in our sample included subspecialty pediatric palliative care in the development, design, or implementation of intervention studies. Research collaborations between NICU/PICU and palliative care teams to develop and test family-centered interventions may be a critical step to strengthen the evidence base for pediatric and neonatal ICU EOL care.

Most interventions relied on educational sessions, simulations, or protocols to improve clinicians’ knowledge about palliative and EOL care. Educational interventions tailored to the NICU/PICU are critical to upholding quality EOL care since most pre-licensure EOL care education focuses on non-pediatric populations, and pediatric ICU mortality is low, meaning NICU and PICU clinicians may not receive much on-the-job EOL training [80,81,82,83]. Simulation studies focused on preparing clinicians for neonatal and pediatric EOL care have demonstrated positive outcomes for clinicians [72,74,84,85]; however, less is known about the ripple effects of such education on patient and family experiences during EOL care, presenting an important area for further evaluation. Additionally, the impact of educational interventions depends on successful integration into complex clinical workflows [86,87,88]. Future studies of EOL care interventions should routinely consider, report, and evaluate implementation facilitators or barriers to help improve scalability and integration into clinical care [86,87].

Mapping intervention goals and outcomes to EOL care gaps reported by parents [13,16,33] revealed important future priorities. Many parents recount issues with how information was shared during their child’s terminal hospitalization [13,16,33]. We identified fourteen studies focused on improving information sharing during clinical care, through integrating palliative care, supporting decision making, improving clinical structures, and enhancing communication. However, outcomes were mostly clinician perspectives or medical record data, rather than family reports. Few interventions focused on enhancing parental role empowerment [41,66], despite recognized importance by bereaved parents [33,89]. Prior qualitative findings retrospectively articulate how clinicians can effectively partner with parents during serious illness [90] and EOL care [4,12,89]. Future research should aim to (1) engage families in designing interventions, (2) prospectively examine family role empowerment during the uncertainty of PICU EOL care, and (3) evaluate family-driven interventions and outcomes.

## 5. Strengths and Limitations

### 5.1. Strengths and Limitations of Included Studies

Most screened articles were excluded for describing EOL care without implementing an intervention (Appendix A). No randomized trials met our inclusion criteria; most studies used quantitative non-randomized or descriptive designs, often without control groups, and in over 25% of cases, without detailed demographic data. These study features limit generalizability or transferability to other populations outside the study context. While randomized controlled trials may be challenging to conduct during intensive EOL care, alternative designs, such as pragmatic trials, cluster and hybrid effectiveness trials, or causal approaches such as interrupted time series analysis may be feasible. Additionally, few studies engaged families in intervention design or study procedures.

### 5.2. Strengths and Limitations of Review

This scoping review followed a comprehensive, peer-reviewed search strategy and rigorous methodological standards. However, our findings should be interpreted considering several limitations. As with all reviews, our findings are limited due to the choice of databases searched and due to publication bias, which limits our analysis to interventions that have been evaluated and reported in the published literature. The search terms we used were comprehensive and corresponded with leading causes of mortality but may not have captured all causes of death. We limited our sample to interventions conducted in PICUs and NICUs using interprofessional interventions and contained the word nurse in the report. Intervention studies conducted in other settings, such as pediatric oncology, or that did not use an interprofessional approach, could have applicable elements. We limited our focus to last days of life; studies focused on earlier serious illness communication, advanced care planning, or shared decision making could be informative to designing EOL interventions. This scoping review did not aim to compare or evaluate intervention effects, as a meta-analysis was outside of the scope. Our approach and findings may therefore provide a framework to inform future systematic reviews, meta-analyses, and/or realist reviews.

## 6. Conclusions

Empirically evaluated EOL interventions in pediatric and neonatal intensive care units are scarce. Most published EOL care intervention studies focus on improving clinician knowledge of palliative and EOL care using educational approaches or clinical practice changes evaluated with pre-post designs. Few interventions target parent-reported gaps in EOL care, especially parental role empowerment, include families in intervention development, use designs that account for selection bias or confounding factors, or consider implementation barriers/facilitators. These are important areas of future research to strengthen EOL care.

## Figures and Tables

**Figure 1 children-12-01485-f001:**
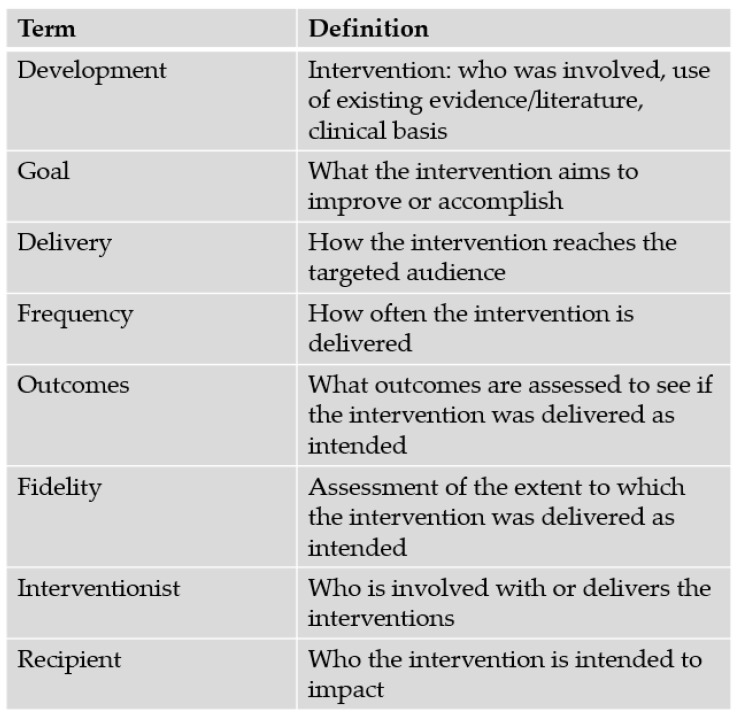
Extracted intervention elements. Alt text: A text box containing definitions of intervention elements.

**Figure 2 children-12-01485-f002:**
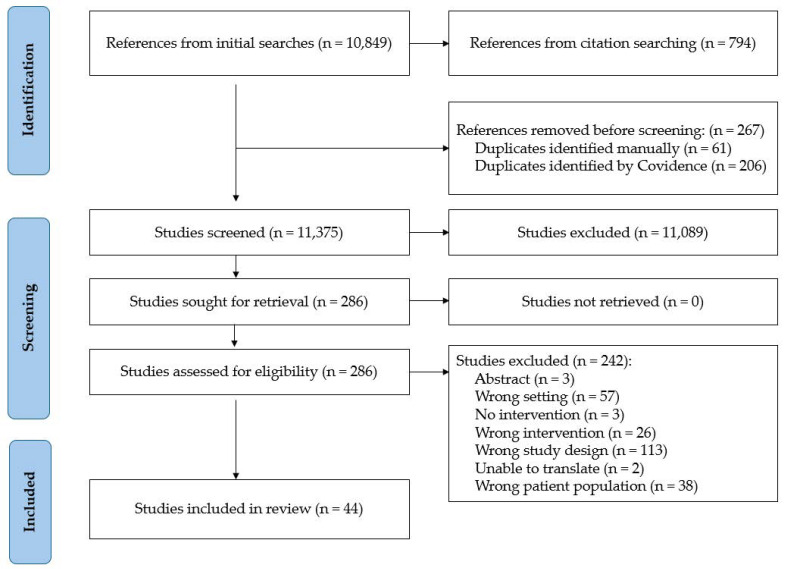
Preferred Reporting Items for Systematic Reviews and Meta-Analyses Flowchart. Alt text: A flowchart of review screening and study selection procedures. Arrows indicate studies that proceeded from one phase to the next.

**Figure 3 children-12-01485-f003:**
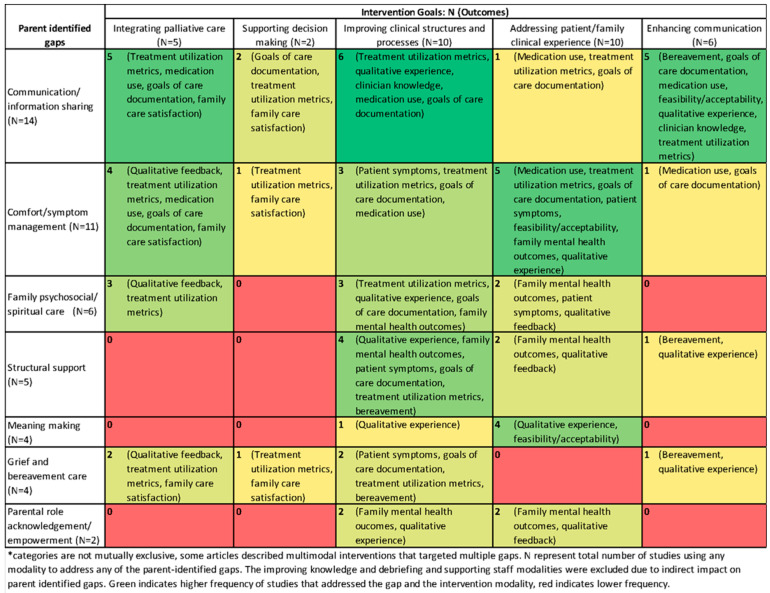
Heat Map of Intervention Goals and Outcomes vs. Parent Identified Gaps in End-of-Life Care. Green indicates more studies addressing a gap with a modality; red indicates fewer. Alt text: Heat map showing how often different intervention types addressed parent-identified gaps in pediatric end-of-life care. Green indicates more studies addressing a gap with a modality; red indicates fewer. * categories are not mutually exclusive, some articles described multimodal interventions that targeted multiple gaps.

**Table 1 children-12-01485-t001:** Aggregate Study Characteristics.

Category	Variable	n (%)
Total		44
Study design	Mixed Methods	3 (7%)
	Other ^1^	8 (18%)
	Qualitative	4 (9%)
	Quantitative descriptive	12 (27%)
	Quantitative non-randomized	17 (39%)
Unit type ^2^	General PICU	10 (23%)
	PCICU	4 (9%)
	NICU	28 (64%)
	Other unit type	2 (5%)
Palliative care domains *	Physical	34 (77%)
	Physical alone	5 (11%)
	Emotional/psychological	38 (86%)
	Emotional alone	1 (2%)
	Spiritual Domain	25 (57%)
	Spiritual alone	0
	Social Domain	30 (68%)
	Social Alone	0
	Multiple	38 (86%)
Interventionist Role ^3^	Interprofessional team + family	2 (5%)
	Research team	4 (10%)
	Supportive care consultants	7 (17%)
	Physician	6 (14%)
	Nurse	3 (7%)
	Interprofessional team	10 (24%)
	ICU clinical team	5 (12%)
	Palliative care team	2 (5%)
	External education team	3 (7%)
	Nurse(s) involved	18 (41%)
Sample *	Patients	14 (32%)
	Parent/family	10 (23%)
	Clinicians	28 (64%)

^1^ Other designs included implementation, quality improvement, informal evaluation, and multimethod. ^2^ Other unit types include interventions where all pediatric units were eligible and over half of intervention participants were in intensive care settings. ^3^ Two interventionist roles were unreported or unclear. * indicates categories that are not mutually exclusive.

## Data Availability

The original contributions presented in this study are included in the article/Appendix A. Further inquiries can be directed to the corresponding author.

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
