# Peer review of "Improving Intensive End-of-Life Care for Infants and Children: A Scoping Review of Intervention Elements"

_children, 2025, doi:10.3390/children12111485_

Round 1

Reviewer 1 Report

Comments and Suggestions for Authors

This is a review of the article entitled “Improving Intensive End-of-Life Care for Infants and Children: A Scoping Review of Intervention Elements.” Thank you for the opportunity to read the article. This article is a scoping review of end-of-life care interventions in which nurses are involved, focusing on the last days of life of infants and children admitted to intensive care units. It was written based on Arksey and O’Malley’s method and has a protocol in OSF which is considered one of the strengths of the study. Authors should pay attention to the following points:

  1. The article has been published as a preprint on October 6, 2025 (Pre-prints.org), which should be mentioned in the methods section with the DOI code.
  2. If there are items related to protocol changes, they should be mentioned in the protocol amendment in the first section of the methods.
  3. The authors used Arksey and O’Malley’s method, the steps of this method should be written in the methods section, and it should be mentioned that step 6 was not performed.
  4. It seems that the authors used PRISMA-ScR checklist (Preferred Reporting Items for Systematic reviews and Meta-Analyses extension for Scoping Reviews) which should be corrected in the text of the article in the methods section.
  5. According to Figure 2, 242 studies were removed from the study after reading the full text. These studies should be specified in a table in the supplementary section of the article with the name of the study and the authors and the reason for deletion.
  6. Figure 2- The second column of the PRISMA diagram: “references from other sources.” Do the authors mean grey literature? These sources should be identified in the text or in the diagram.
  7. Since “PubMed” does not only include Medline information, why did the authors not perform their search on this platform, also?
  8. Since the authors chose interventional studies, why didn't they search clinical trial websites, such as WHO trials (ICTRP) and so on?
  9. Introduction: “We therefore conducted a scoping review to evaluate interventions for children nearing EOL in the PICU.” The authors should note this throughout the article: They collected studies data not only in the PICUs but also in the neonatal intensive care units (NICUs).
  10. Supplementary File- Table of evidence: Please specify the name of the country and publication year in the table.
  11. Selection criteria: “We sought to include interventions that meaningfully involved nurses in their implementation and impact.” But, discussion is a little bit misleading: “While 41% of intervention teams included nurses, only 3 (%) studies evaluated nurse-led interventions, and most interventions did not impact routine nursing practice.” It should be clear here and throughout the article that nurses were involved in various ways in all interventions.
  12. Study rigor: “Many studies (15/44; 34%) did not report sample demographic data.” Please revise the sentence, this is not “many studies.” It can be revised to: “Only 66% of studies reported demographic data of the sample.”

Author Response

Response to Reviewer 1 Comments

Summary

We appreciate the detailed comments from Reviewer 1. Their insights have helped improve the manuscript, as detailed below.

Point-by-point response to Comments and Suggestions for Authors

Overall comments: This is a review of the article entitled “Improving Intensive End-of-Life Care for Infants and Children: A Scoping Review of Intervention Elements.” Thank you for the opportunity to read the article. This article is a scoping review of end-of-life care interventions in which nurses are involved, focusing on the last days of life of infants and children admitted to intensive care units. It was written based on Arksey and O’Malley’s method and has a protocol in OSF which is considered one of the strengths of the study. Authors should pay attention to the following points:

Overall response: Thank you for the detailed read of our manuscript and supplying opportunities to improve.

Comments 1: The article has been published as a preprint on October 6, 2025 (Pre-prints.org), which should be mentioned in the methods section with the DOI code.

Response 1: I have added information about the preprint (including the DOI) to the methods section.

The manuscript was published on a pre-print server on October 6th, 2025 (https://doi.org/10.20944/preprints202510.0396.v1).

Comments 2: If there are items related to protocol changes, they should be mentioned in the protocol amendment in the first section of the methods.

Response 2: Thank you. I have revised the methods section to include protocol changes:

We made a minor protocol amendments, specifically to the synthesis and analysis phase, as needed based on the content of included articles.  

Comments 3: The authors used Arksey and O’Malley’s method, the steps of this method should be written in the methods section, and it should be mentioned that step 6 was not performed.

Response 3: I have updated the methods section to include the steps of Arksey and O’Malley’s review approach and that step 6 was not performed.

This scoping review [20] followed Arksey and O’Malley’s method [21], including 1) identifying the research question (section 2.1), 2) identifying relevant studies (section 2.2), 3) study selection (section 2.3), 4) charting the data (sections 2.5 and 2.6), and 5) collating, summarizing, and reporting the results (section 2.6 and section 3).  We did not incorporate a patient and family engagement step as proposed by Levac et al [22].

Comments 4: It seems that the authors used PRISMA-ScR checklist (Preferred Reporting Items for Systematic reviews and Meta-Analyses extension for Scoping Reviews) which should be corrected in the text of the article in the methods section.

Response 4: I revised the methods to describe the PRISMA-ScR and included.

We used the Preferred Reporting Items for Systematic Reviews and Meta-Analyses (PRISMA-ScR) guidelines Extension for Scoping Reviews [23].

Comments 5: According to Figure 2, 242 studies were removed from the study after reading the full text. These studies should be specified in a table in the supplementary section of the article with the name of the study and the authors and the reason for deletion.

Response 5: We created a second supplement containing more details about excluded studies.

Comments 6: Figure 2- The second column of the PRISMA diagram: “references from other sources.” Do the authors mean grey literature? These sources should be identified in the text or in the diagram.

Response 6: I updated the PRISMA flowchart to indicate that these references were all identified from searching the references of included papers.

Comments 7: Since “PubMed” does not only include Medline information, why did the authors not perform their search on this platform, also?

Response 7: While the peer reviewer raises a good question, studies have shown that these databases produce comparable results, so one was chosen rather than search both. (See Katchamart, W., Faulkner, A., Feldman, B., Tomlinson, G., & Bombardier, C. (2011). PubMed had a higher sensitivity than Ovid-MEDLINE in the search for systematic reviews. Journal of Clinical Epidemiology., 64(7), 805–807. https://doi.org/10.1016/j.jclinepi.2010.06.004; & Stillwell, S. B., & Scott, J. G. (2020). Sensitive Versus Specific Search Strategy to Answer Clinical Questions. JNE. Journal of Nursing Education., 59(1), 22–25. https://doi.org/10.3928/01484834-20191223-05.

Based on this feedback, though, we also updated the limitations section:

As with all reviews, our findings are limited due to the choice of databases searched, our findings and due to publication bias, which limits our analysis to interventions that have been evaluated and reported in published literature.

Comments 8: Since the authors chose interventional studies, why didn't they search clinical trial websites, such as WHO trials (ICTRP) and so on?

Response 8: We chose not to include information sources other than published reports because we wanted to evaluate efforts to change clinical practice rather than research methods/findings alone. The Limitations section has been amended accordingly:

As with all reviews, our findings are limited due to the choice of databases searched, our findings and due to publication bias, which limits our analysis to interventions that have been evaluated and reported in published literature.

Comments 9: Introduction: “We therefore conducted a scoping review to evaluate interventions for children nearing EOL in the PICU.” The authors should note this throughout the article: They collected studies data not only in the PICUs but also in the neonatal intensive care units (NICUs).

Response 9: Thank you. I have updated the language throughout to be consistent with including both NICU and PICU studies.

Comments 10: Supplementary File- Table of evidence: Please specify the name of the country and publication year in the table.

Response 10: Thank you. We confirm that this information is now in the table of evidence. The TOE contains the publication year in column 2, and country in column 3

Comments 11: Selection criteria: “We sought to include interventions that meaningfully involved nurses in their implementation and impact.” But, discussion is a little bit misleading: “While 41% of intervention teams included nurses, only 3 (%) studies evaluated nurse-led interventions, and most interventions did not impact routine nursing practice.” It should be clear here and throughout the article that nurses were involved in various ways in all interventions.

Response 11: We appreciate this comment very much and have updated the selection criteria rationale to this effect.

Selection criteria: To capture interprofessional interventions, articles with the search terms in the title, abstract, or subheadings and the word “nurse” within the full text were included. This allowed us to evaluate interventions that meaningfully involved nurses in their implementation and impact in a variety of ways.

Discussion: Over 40% of intervention teams involved nurses as part of the research team or in intervention design. However, just 3 (7%) studies evaluated nurse-led interventions that changed routine nursing practice, such as through hands on care[41], assessment procedures[46] or bereavement interventions and follow up[47].

Comments 12: Study rigor: “Many studies (15/44; 34%) did not report sample demographic data.” Please revise the sentence, this is not “many studies.” It can be revised to: “Only 66% of studies reported demographic data of the sample.”

Response 12: Thank you for this suggestion. I made this change and updated the limitation section. The edited sentence in the findings reads:

Two-thirds, 66% (29/44), of studies reported demographic data of their sample.

The edited sentence in the limitations reads:

No randomized trials met our inclusion criteria; most studies used quantitative non-randomized or descriptive designs, often without control groups, and in over 25% of cases without detailed demographic data.

Reviewer 2 Report

Comments and Suggestions for Authors

This paper makes a valuable descriptive and methodological contribution by consolidating the fragmented evidence on pediatric EOL interventions. Some revisions could strengthen the work in my opinion.

Introduction: Explicitly link the synthesis to one or more conceptual or implementation frameworks (e.g., CFIR, Family-Centered Care, or the MRC Framework). This would elevate the review from cataloging to interpretation. After the aim, clarify how your mapping of interventions informs theoretical understanding of how EOL care elements function across settings (significance of the study).

Discussion. Reorganize the Discussion to follow a logical flow: (1) synthesis of key elements, (2) nursing implications, (3) implementation considerations, (4) research gaps. In general, move beyond descriptive categories by analyzing patterns of effectiveness or mechanisms (e.g., which intervention elements appear to reduce decisional distress or improve family communication). Identify which family-reported needs remain least addressed (e.g., bereavement support, sibling care, role negotiation) and offer 2–3 concrete research priorities for future interventional or implementation studies. In limitations, explicitly acknowledge that a scoping review cannot assess intervention quality or effectiveness and emphasize that the purpose is to inform future systematic or realist reviews rather than draw evaluative conclusions.

Author Response

Response to Reviewer 2 Comments

Summary

We extend our gratitude to Reviewer 2 for their suggestions to improve the clarity of the introduction and discussion in the report. Details reported below.  

Point-by-point response to Comments and Suggestions for Authors

Overall comments: This paper makes a valuable descriptive and methodological contribution by consolidating the fragmented evidence on pediatric EOL interventions. Some revisions could strengthen the work in my opinion.

Overall Response: Thank you for identifying opportunities to revise our manuscript.

Comments 1: Introduction: Explicitly link the synthesis to one or more conceptual or implementation frameworks (e.g., CFIR, Family-Centered Care, or the MRC Framework). This would elevate the review from cataloging to interpretation. After the aim, clarify how your mapping of interventions informs theoretical understanding of how EOL care elements function across settings (significance of the study).

Response 1: The reviewer raises an important point. However, because a meta-analysis was outside the scope of this review, we cannot interpret how different factors influenced intervention implementation or impact. We retain the language of cataloging intervention and implementation elements, and recommend that future researchers evaluate and interpret how the implementation factors that were frequently discussed by authors cited in our review.

I have added more details about the CFIR, which informed data extraction and analysis to the methods and results.

Here is the updated text from section 2.5 (Data Extraction):

We extracted unit- and hospital-level intervention implementation considerations based on the Consolidated Framework for Implementation Research, including barriers and facilitators within the domains of innovation design, outer setting, inner setting, and individuals [26,27].

Below is the updated text from the results:

Hospital-level (outer setting) implementation facilitators included committed intervention champions (7/44; 16%), support for required time/resources (4/44; 9%) [59,68,70,72], established hospital EOL, palliative, or bereavement programming (7/44; 16%). Barriers included lack of programming (4/44; 9%) [50,65,69,70]. Many studies (26/44; 59%) did not report about hospital-level implementation factors.

Unit-level (inner setting) implementation facilitators included high volume (12/44; 27%); existing family-centered, palliative, or EOL care infrastructure (8/44; 18%). Individual level facilitators included invested individuals (8/44; 18%); and staff enthusiasm (2/44; 5%) [44,57]. Unit-level (inner setting) implementation barriers included technology or procedural issues (10/44; 23%); scheduling (10/44; 23%); lack of palliative or EOL care infrastructure (3/44; 7%) [61,70,71]. Individual hesitations (2/44; 5%) [41,60] also functioned as barriers, for example, nurses who used an oral or transmucosal comfort medication protocol in Drolet et al.’s study reported discomfort interrupting families’ privacy [41].

And here is the updated language from paragraph 3 of the discussion section:

Future studies of EOL care interventions should routinely consider, report, and evaluate implementation facilitators or barriers to help improve scalability and integration into clinical care [38,39].

Comments 2: Discussion. Reorganize the Discussion to follow a logical flow: (1) synthesis of key elements, (2) nursing implications, (3) implementation considerations, (4) research gaps. In general, move beyond descriptive categories by analyzing patterns of effectiveness or mechanisms (e.g., which intervention elements appear to reduce decisional distress or improve family communication). Identify which family-reported needs remain least addressed (e.g., bereavement support, sibling care, role negotiation) and offer 2–3 concrete research priorities for future interventional or implementation studies. In limitations, explicitly acknowledge that a scoping review cannot assess intervention quality or effectiveness and emphasize that the purpose is to inform future systematic or realist reviews rather than draw evaluative conclusions.

Response 2: The reviewer raises a helpful suggestion. This scoping review did not include a meta-analysis to examine effectiveness of different interventions, thus the discussion cannot speak to patterns of effectiveness. Paragraph 1 summarizes key findings. I have reorganized the rest of the discussion with these recommendations, including adding a list of specific research priorities. I am pasting below the major changes to the content of the discussion below:

Paragraph 2 of the discussion now focuses on nursing implications:

NICU and PICU nurses are uniquely positioned to help assess, support, and advocate for families’ dynamic needs during EOL. Over 40% of intervention teams involved nurses as part of the research team or in intervention design. However, just 3 (7%) studies evaluated nurse-led interventions that changed routine nursing practice, such as through hands on care [42], assessment procedures [47] or bereavement interventions and follow up [48]. Given nurses’ uniquely close involvement with children and families during EOL, future interventions should consider leveraging nursing workflows, such as nurse communication and partnership with parents, to enhance EOL care.

Paragraph 3 of the discussion now focuses on collaboration between ICU teams and palliative care teams:

Many studies support early palliative care during a child’s serious illness [37,38]. However, few studies in our sample included subspecialty pediatric palliative care in the development, design, or implementation of intervention studies. Research collaborations between NICU/PICU and palliative care teams to develop and test family-centered interventions may be a critical step to strengthen the evidence base for PICU EOL care.

Paragraph 4 of the discussion now focuses on implementation considerations and reads:

Most interventions relied on educational sessions, simulations, or protocols to improve clinicians’ knowledge about palliative and EOL care. Educational interventions tailored to the NICU/PICU are critical to upholding quality EOL care since most pre-licensure EOL care education focuses on non-pediatric populations, and pediatric ICU mortality is low, meaning NICU and PICU clinicians may not receive much on-the-job EOL training [80–83]. Simulation studies focused on preparing clinicians for neonatal and pediatric EOL care have demonstrated positive outcomes for clinicians[72,74,74,74,84,85]; however, less is known about the ripple effects of such education on patient and family experiences during EOL care, presenting an important area for further evaluation. Additionally, the impact of educational interventions depends on successful integration into complex clinical workflows [86–88]. Future studies of EOL care interventions should routinely consider, report, and evaluate implementation facilitators or barriers to help improve scalability and integration into clinical care [86,87]. 

Paragraph 5 of the discussion now focuses on future research:

Mapping intervention goals and outcomes to EOL care gaps reported by parents [13,16,33] revealed important future priorities. Many parents recount issues with how information was shared during their child’s terminal hospitalization [13,16,33]. We identified fourteen studies focused on improving information sharing during clinical care, through integrating palliative care, supporting decision making, improving clinical structures, and enhancing communication. However, outcomes were mostly clinician perspectives or medical record data, rather than family reports. Few interventions focused on enhancing parental role empowerment, [34,46] despite perceived importance by bereaved parents [33,47]. Prior qualitative findings retrospectively articulate how clinicians can effectively partner with parents during serious illness [48] and EOL care [4,12,47]. Future research should aim to 1) engage families in designing interventions, 2) prospectively examine family role empowerment during the uncertainty of PICU EOL care, and 3) evaluate of family-driven interventions and outcomes. 

I added the following to the limitations:

This scoping review did not aim to compare or evaluate intervention effects, as a meta-analysis was outside of the scope. Our approach and findings may therefore provide a framework to inform future systematic reviews, meta-analyses, and/or realist reviews.

Reviewer 3 Report

Comments and Suggestions for Authors

Thank you very much for the opportunity to review your paper: 'Improving Intensive End-of-Life Care for Infants and Children: A Scoping Review of Intervention Elements'. End-of-life care in general has been receiving increasing attention in scientific literature recently. The focus on infants and children is therefore certainly relevant within the current landscape. Furthermore, it is courageous to highlight this topic. Please find some suggestions below:

Highlights

  • L22 isn't this obvious? It is the aim of the paper.

Abstract

  • NICU is included in the results but not mentioned in the background. Please briefly mention this as well, because NICU and PICU are two different target groups.
  • Keywords: think about including NICU and PICU as well. I suspect the article will have more reach by including this.

Introduction

  • Since NICU is also discussed later in the article, this concept needs to be introduced.
  • Explain the reasoning more clearly. Guide the reader in interpreting the statements. E.g. why is a multidisciplinary, team-based approach crucial?
  • L61 Is this sentence in the right place?
  • L79 Here you indicate that it only concerns children in PICU, but earlier you also mentioned infants and NICU.
  • The information in the introduction is correct but too limited. Explain the concepts in more detail (e.g. EOL). Also, ensure a logical structure in which you take the reader through your rationale from problem definition to research question.

Methods

  • Clearly written method that enables the reader to repeat the study or understand how it was conducted.

Results

  • Full display of results.

Discussion

  • Be more critical in the discussion. The results and their implications are currently being presented. However, I feel that a critical assessment of existing literature and recommendations are lacking. For example, simulation is mentioned but not elaborated upon. There is literature on simulation and euthanasia or simulation and palliative care. Although not in a NICU or PICU context, this could be a good suggestion.

Conclusion

  • Well done

Author Response

Response to Reviewer 3 Comments

Summary

We’d like to thank Reviewer 3 for identifying opportunities to clarify and strengthen the reporting of our scoping review.  

Point-by-point response to Comments and Suggestions for Authors

Overall comments: Thank you very much for the opportunity to review your paper: 'Improving Intensive End-of-Life Care for Infants and Children: A Scoping Review of Intervention Elements'. End-of-life care in general has been receiving increasing attention in scientific literature recently. The focus on infants and children is therefore certainly relevant within the current landscape. Furthermore, it is courageous to highlight this topic. Please find some suggestions below:

Overall response: We sincerely appreciate your recognition of the importance of this topic, thank you. Our detailed responses are below.

Comments 1: Highlights - L22 isn't this obvious? It is the aim of the paper.

Response 1: Thank you for raising this point. I have updated the highlights to read:

·       Most interventions target clinician knowledge as a primary outcome; whereas fewer interventions targeted family outcomes.

·       Few interventions utilized nursing workflows to improve end-of-life care in pediatric and neonatal critical care settings.

Comments 2: Abstract - NICU is included in the results but not mentioned in the background. Please briefly mention this as well, because NICU and PICU are two different target groups.

Keywords: think about including NICU and PICU as well. I suspect the article will have more reach by including this.

Response 2: Thank you for pointing out this inconsistency in reporting and opportunity to expand the reach of the work. We have added NICU to the abstract text and keywords.  

Comments 3: Introduction - Since NICU is also discussed later in the article, this concept needs to be introduced. - Explain the reasoning more clearly. Guide the reader in interpreting the statements. E.g. why is a multidisciplinary, team-based approach crucial?

Response 3: I made updates to the introduction based on these suggestions and those from Reviewer 1 (Comment 9). I have also added NICU/PICU throughout the manuscript to ensure it is clear both of these settings are clearly represented. The below sentence is now included in paragraph 1 of the introduction:

Optimal end-of-life for infants and children care includes a multidisciplinary, team-based approach that considers physical symptoms, family spiritual and emotional needs, and meaning-making opportunities [7,11,12].

Comments 4: L61 Is this sentence in the right place?

Response 4: I removed this sentence, and edited the next sentence:

As the clinicians most frequently at the bedside, nurses are well-positioned to assess, evaluate, and connect families with the support they need during EOL.

Comments 5: L79 Here you indicate that it only concerns children in PICU, but earlier you also mentioned infants and NICU.

Response 5: Based on this comment and suggestions from reviewer 2, NICU is now included throughout the manuscript. The sentence now reads:

Systematically cataloging existing interventions is necessary to ensure all dying children in the NICU/PICU receive compassionate, evidence-based care from well-trained clinicians. We therefore conducted a scoping review to evaluate interventions for children nearing EOL in the NICU/PICU.

Comments 6: The information in the introduction is correct but too limited. Explain the concepts in more detail (e.g. EOL). Also, ensure a logical structure in which you take the reader through your rationale from problem definition to research question.

Response 6: Thank you for this advice. I updated the flow of the introduction based on these suggestions. I also added a definition for EOL and stronger rationale for the research question. The first two paragraphs of the into now read:

 A child’s death is life-altering, leading to life-long grief for their families [1–7]. Many children and families navigate end-of-life (EOL), the last days or hours of an illness or injury, in a neonatal or pediatric intensive care unit (NICU, PICU) [8–10]. Optimal end-of-life for infants and children care includes a multidisciplinary, team-based approach that considers physical symptoms, family spiritual and emotional needs, and meaning-making opportunities [7,11,12]. As the clinicians most frequently at the bedside, nurses are well-positioned to assess, evaluate, and connect families with the support they need during EOL. Interprofessional EOL care that fully leverages nurses’ roles is therefore critical to supporting families and minimizing adverse grief outcomes. 

Optimizing interprofessional EOL care for infants and children in the ICU setting requires an evidence base of rigorously tested interventions. Generally low pediatric mortality rates mean that clinicians may rarely apply palliative and EOL skills learned during pre-licensure training, instead learning on the job. Studies of family and clinician experiences illuminate several gaps in EOL care that exacerbate parental distress [13–16]. These include fragmented communication, strained relationships within and between clinical teams and families, and limited structural support for families and clinicians, among others [13–16]. Supportive interventions that can be readily integrated into routine care could help address these known barriers to high-quality EOL care.  While many interventions focused on improving EOL care have been developed, a systematic understanding of what elements interventions include and which parent-identified gaps they target is lacking.

Comments 7: Methods - Clearly written method that enables the reader to repeat the study or understand how it was conducted.

Response 7: Thank you for this comment!

Comments 8: Results - Full display of results.

Response 8: Thank you for this comment.

Comments 9: Be more critical in the discussion. The results and their implications are currently being presented. However, I feel that a critical assessment of existing literature and recommendations are lacking. For example, simulation is mentioned but not elaborated upon. There is literature on simulation and euthanasia or simulation and palliative care. Although not in a NICU or PICU context, this could be a good suggestion.

Response 9: Thank you for pointing this out. Based on this feedback and suggestions from Reviewer 2, I have added more critical assessment and recommendations to the discussion, including:

Paragraph 2: NICU and PICU nurses are uniquely positioned to help assess, support, and advocate for families’ dynamic needs during EOL. Over 40% of intervention teams involved nurses as part of the research team or in intervention design. However, just 3 (7%) studies evaluated nurse-led interventions that changed routine nursing practice, such as through hands on care[66], assessment procedures[60] or bereavement interventions and follow up[70]. Given nurses’ uniquely close involvement with children and families during EOL, future interventions should consider leveraging nursing workflows, such as nurse communication and partnership with parents, to enhance EOL care.

Paragraph 4: Most interventions relied on educational sessions, simulations, or protocols to improve clinicians’ knowledge about palliative and EOL care. Educational interventions tailored to the NICU/PICU are critical to upholding quality EOL care since most pre-licensure EOL care education focuses on non-pediatric populations, and pediatric ICU mortality is low, meaning NICU and PICU clinicians may not receive much on-the-job EOL training [80–83]. Simulation studies focused on preparing clinicians for neonatal and pediatric EOL care have demonstrated positive outcomes for clinicians[72,74,74,74,84,85]; however, less is known about the ripple effects of such education on patient and family experiences during EOL care, presenting an important area for further evaluation. Additionally, the impact of educational interventions depends on successful integration into complex clinical workflows [86–88]. Future studies of EOL care interventions should routinely consider, report, and evaluate implementation facilitators or barriers to help improve scalability and integration into clinical care [86,87]. 

Paragraph 5: Mapping intervention goals and outcomes to EOL care gaps reported by parents [13,16,33] revealed important future priorities. Many parents recount issues with how information was shared during their child’s terminal hospitalization [13,16,33]. We identified fourteen studies focused on improving information sharing during clinical care, through integrating palliative care, supporting decision making, improving clinical structures, and enhancing communication. However, outcomes were mostly clinician perspectives or medical record data, rather than family reports. Few interventions focused on enhancing parental role empowerment, [41,66] despite perceived importance by bereaved parents [33,89]. Prior qualitative findings retrospectively articulate how clinicians can effectively partner with parents during serious illness [90] and EOL care [4,12,89]. Future research should aim to 1) engage families in designing interventions, 2) prospectively examine family role empowerment during the uncertainty of PICU EOL care, and 3) evaluate of family-driven interventions and outcomes. 

Comments 10: Conclusion - Well done

Response 10: Thank you.